# Learning Inertial Odometry for Dynamic Legged Robot State Estimation

**Russell Buchanan**
Oxford Robotics Institute
University of Oxford
russell@robots.ox.ac.uk

**Marco Camurri**
Oxford Robotics Institute
University of Oxford
mcamurri@robots.ox.ac.uk

**Frank Dellaert**
Institute for Robotics
and Intelligent Machines
Georgia Institute of Technology
dellaert@cc.gatech.edu

**Maurice Fallon**
Oxford Robotics Institute
University of Oxford
mfallon@robots.ox.ac.uk

**Abstract:** This paper introduces a novel proprioceptive state estimator for legged robots based on a learned displacement measurement from IMU data. Recent research in pedestrian tracking has shown that motion can be inferred from inertial data using convolutional neural networks. A learned inertial displacement measurement can improve state estimation in challenging scenarios where leg odometry is unreliable, such as slipping and compressible terrains. Our work learns to estimate a displacement measurement from IMU data which is then fused with traditional leg odometry. Our approach greatly reduces the drift of proprioceptive state estimation, which is critical for legged robots deployed in vision and lidar denied environments such as foggy sewers or dusty mines. We compared results from an EKF and an incremental fixed-lag factor graph estimator using data from several real robot experiments crossing challenging terrains. Our results show a reduction of relative pose error by $37\%$ in challenging scenarios when compared to a traditional kinematic-inertial estimator without learned measurement. We also demonstrate a $22\%$ reduction in error when used with vision systems in visually degraded environments such as an underground mine.

**Keywords:** Legged Robots, Inertial Navigation, Deep Neural Networks

## 1 Introduction

Recent advances in legged robot locomotion have motivated the use of quadruped robots in dull and dirty industrial applications. To perform these tasks autonomously, accurate state estimation with limited drift over long periods of time is fundamental. However, operating in challenging environments such as mines and sewers, dust particles and vapor can severely degrade camera and lidar systems [1] (Fig. 1). Proprioceptive methods can work in these environments, as they fuse information only from joint kinematics and Inertial Measurement Unit (IMU) data [2, 3, 4]. However, slip events or deformation of the robot's foot or terrain introduce non-Gaussian error that accumulates over time and leads to unbounded drift.

Separately, recent advances in pedestrian tracking using only IMU data with machine learning have shown promising results [5, 6]. The key insight of these works is that it is possible to learn a *motion prior* from inertial data which could be robust to changing biases. We propose a deep neural network capable of learning a motion prior on the locomotion of a legged robot for any terrain and in challenging conditions such as slipping. The network takes a buffer of IMU data and outputs a displacement measurement and covariance which is fused with kinematic information. We demonstrate the application of this measurement in both filtering and factor-graph optimization contexts to track the motion of an ANYbotics ANYmal quadrupedal robot and compare results to other learning-based inertial odometry systems [5].

5th Conference on Robot Learning (CoRL 2021), London, UK.

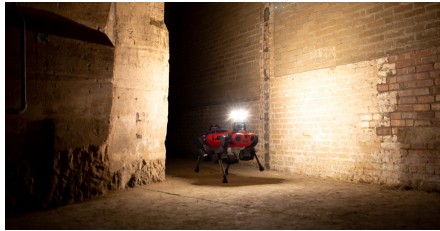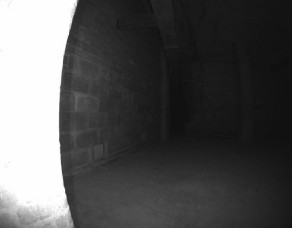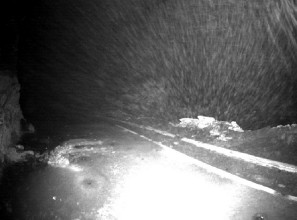

Figure 1: **Left:** The ANYmal C100 quadrupedal robot exploring a limestone mine. **Right:** Images from the on-board camera which demonstrate the vision challenges in such an environment.

The contributions can be listed as follows: 1) First kinematics-inertial estimator based on learned IMU displacement measurements; 2) Integration with both a filter-based estimator and a factor graph, merging concepts from machine learning and model based methods; 3) Extensive testing on a quadruped robot, including application to field experiments of the robot navigating a mine.

## 2 Related Work

In this section, we briefly summarize relevant work on proprioceptive state estimation for legged robots (Section 2.1) and related research in learned inertial state estimation (Section 2.2).

### 2.1 Kinematic-Inertial State Estimation in Legged Robots

Lin et al. [7] presented the first state estimator for legged robots which fused IMU and kinematic data in an Extended Kalman Filter (EKF). Their method was highly dependent on the assumption that their hexapedal robot had always three feet in contact with the ground. Bloesch et al. [2] introduced a new filtering approach called the Two-State Implicit Filter (TSIF). This method employs residual-based modeling of the available information, eliminating the need for an explicit process model. The default state estimator for the ANYmal robot, which we use in our experiments, is based on TSIF.

More recently, Pronto [8] has been presented as an EKF-based state estimator with the capacity to fuse pose corrections from exteroceptive sensing. This is done by maintaining a history of states, measurements and covariances, and applying corrections to the past trajectory asynchronously.

Factor graphs have also been used for legged robot state estimation. Hartley et al. [9] proposed a hybrid preintegrated contact factor so that multiple foot steps or measurements could be combined into a single factor, reducing the number of nodes in the graph. Wisth et al. [10] used factor graphs to fuse visual information with inertial and kinematics. In [11], the same authors improved on the factor graph formulation by adding feature tracking-based lidar measurements.

### 2.2 Learned Inertial Navigation

Recent research in inertial navigation systems (INS) has applied machine learning to infer motion directly from IMU data — typically focused on tracking a handheld mobile phone. These methods have shown impressive performance when compared to traditional inertial-only pedestrian tracking methods. However, these methods are susceptible to failure if applied outside of the training domain (e.g., a network trained with pedestrian motion would not generalize to a person on a bicycle).

Chen et al. [6] introduced IoNet, the first data driven INS which estimates odometry using deep recurrent neural networks (RNNs). The network was trained to relate buffered data from an IMU into 2D displacements and orientation changes. Yan et al. [12] proposed RoNIN, a network architecture which similarly infers 2D velocity and orientation from raw IMU data.

The first approach estimating 3D motion was proposed by Liu et al. [5], who presented the TLIO filtering framework including learned motion estimates from batches of IMU signals. An EKF was used to filter the full 6 DoF state and process updates were performed by traditional IMU integration. They trained a network with $40\,\mathrm{h}$ of pedestrian data to output displacement and covariance from the buffered IMU data. These measurements were provided to the EKF as relative position measurements and allowed the filter to correct for biases online. Their approach showed impressive results for IMU only navigation but was unable to integrate other sensors such as kinematics or vision.

Currently, the application of inertial learning to robotics is limited. Brossard et al. [13] demonstrated IMU-only state estimation specifically for wheeled vehicles, heavily exploiting the constrained motion model. Zhang et al. [14] trained a network to learn the noise parameters of the IMU as opposed to the motion of the subject. Their RNN generates rotation, velocity and acceleration terms with bias removed, however this makes integration with other frameworks such as filtering more difficult.

## 3 Problem Statement

Our objective is to estimate the state of a legged robot equipped with an IMU, joint sensors (encoders, torque sensors) and, optionally, exteroceptive sensors. Following standard conventions, we define a fixed world frame W aligned with gravity, and two frames attached to the robot: the base frame B and the IMU frame I. The transformations between B and I are known. The robot's state at the current time $\boldsymbol{x}(t_k) = \boldsymbol{x}_k$ is defined as follows:

$$\boldsymbol{x}_k = [\mathbf{R}_k, \mathbf{p}_k, \mathbf{v}_k, \mathbf{b}_k^a, \mathbf{b}_k^g] \tag{1}$$

and includes the rotation $\mathbf{R}_{\mathtt{WB}} \in \mathrm{SO}(3)$ in world coordinates, the position ${}_{\mathtt{W}}\mathbf{p}_{\mathtt{WB}} \in \mathbb{R}^3$, the linear velocity ${}_{\mathtt{B}}\mathbf{v}_{\mathtt{WB}}$, and the gyro and accelerometer IMU biases ${}_{\mathtt{I}}\mathbf{b}^g$, ${}_{\mathtt{I}}\mathbf{b}^a$. The IMU measures rotational velocity ${}_{\mathtt{I}}\tilde{\boldsymbol{\omega}}_{\mathtt{WI}} \in \mathbb{R}^3$ and the proper acceleration ${}_{\mathtt{I}}\tilde{\mathbf{a}} \in \mathbb{R}^3$. These are corrupted by zero-mean Gaussian noise $\boldsymbol{\eta}^g, \boldsymbol{\eta}^a$ and by slowly changing biases:

$$\tilde{\boldsymbol{\omega}}(t) = \boldsymbol{\omega}(t) + \mathbf{b}^g(t) + \boldsymbol{\eta}^g(t) \qquad \tilde{\mathbf{a}}(t) = \mathbf{R}^{\mathsf{T}}({}_{\mathtt{W}}\mathbf{a}(t) - {}_{\mathtt{W}}\mathbf{g}) + \mathbf{b}^a(t) + \boldsymbol{\eta}^a(t) \tag{2}$$

where ${}_{\mathtt{W}}\mathbf{g}$ is the gravity vector in world frame. Without constraining measurements, the biases are not observable and simple IMU integration will quickly lead to runaway error.

## 4 Learned Displacement Measurement

In this section, we describe our learning method including network architecture and loss functions. Let $\mathcal{I}_{ij}$ be a sequence of $N$ consecutive IMU measurements collected from time $t_i$ to $t_j$ and gravity aligned. From $\mathcal{I}_{ij}$, we want to predict the robot's linear position increment $\hat{\mathbf{d}}_i \in \mathbb{R}^3$ between $t_i$ and $t_j$. To this end, we define a nonlinear function $\mathcal{F}_\theta$ that will be approximated by a neural network:

$$\mathcal{F}_\theta : (\mathcal{I}_{ij}) \mapsto (\hat{\mathbf{d}}_i, \hat{\mathbf{u}}_i) \tag{3}$$

where $\hat{\mathbf{u}}_i = (\hat{u}_i^x, \hat{u}_i^y, \hat{u}_i^x)$ contains the logarithms of standard deviations from which the covariance of the displacement $\hat{\boldsymbol{\Sigma}}_i^d \in \mathbb{R}^{3\times3}$ can be computed:

$$\hat{\boldsymbol{\Sigma}}_i^d = \mathrm{diag}\left(\exp(2\hat{u}_i^x), \exp(2\hat{u}_i^y), \exp(2\hat{u}_i^z)\right) \tag{4}$$

Following an approach similar to TLIO [5], we approximate $\mathcal{F}_\theta$ with a 1D version of the ResNet18 architecture [15]. We also make use of two loss functions, but in contrast to [5] we use a robust Huber loss kernel to account for outliers in the training set.

### 4.1 Loss Function

For the first 10 epochs, the loss function is defined by the Mean Square Error (MSE):

$$\mathcal{L}(\mathbf{d}, \hat{\mathbf{d}}) = \frac{1}{n} \sum_{k=1}^{n} \left\| \mathbf{d}_k - \hat{\mathbf{d}}_k \right\|^2, \tag{5}$$

where $n$ is the number of training samples, $\hat{\mathbf{d}}$ is the network output, and $\mathbf{d}$ is the ground truth displacement. Once the network has begun to converge, the loss function is changed to a robust modification of the Gaussian Maximum Likelihood (GML), which allows the network to learn uncertainty (as done in [16]) but without being over influenced by outliers. If the MSE loss is not initially used we find the network does not converge. The negative log-loss of the GML is given by:

$$\begin{aligned}
\mathcal{L}(\mathbf{d}, \hat{\boldsymbol{\Sigma}}^d, \hat{\mathbf{d}}) &= \frac{1}{n} \sum_{k=1}^{n} -\log\left( \frac{1}{\sqrt{8\pi \det(\hat{\boldsymbol{\Sigma}}_k^d)}} \exp\left( \mathrm{H}(\mathbf{d}_k - \hat{\mathbf{d}}_k, \hat{\boldsymbol{\Sigma}}_k^d) \right) \right) \\
&= \frac{1}{n} \sum_{k=1}^{n} \left( \frac{1}{2} \log \det(\hat{\boldsymbol{\Sigma}}_k^d) + \mathrm{H}(\mathbf{d}_k - \hat{\mathbf{d}}_k, \hat{\boldsymbol{\Sigma}}_k^d) + \mathrm{const.} \right)
\end{aligned} \tag{6}$$

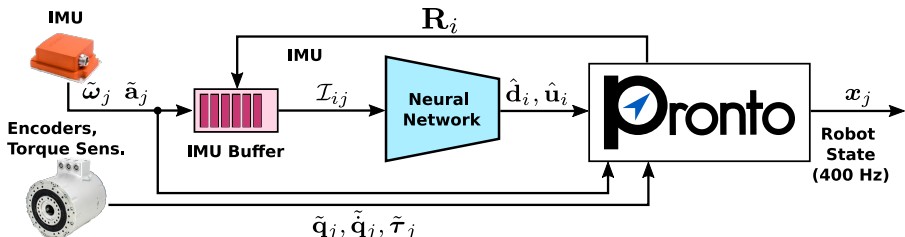

Figure 2: EKF-IKD pipeline: IMU data is integrated in Pronto which provides an estimate of the robot's orientation. The IMU buffer is aligned with gravity and provided as input to the neural network. Displacement measurements and covariance are fused back into Pronto.

where $\text{H}(\boldsymbol{\epsilon}_k, \hat{\boldsymbol{\Sigma}}_k^d)$ is the sum of the Huber loss $\text{h}(\cdot, \cdot)$ applied to each axis of $\boldsymbol{\epsilon}_k = \mathbf{d}_k - \hat{\mathbf{d}}_k$:

$$\text{H}(\boldsymbol{\epsilon}_k, \boldsymbol{\Sigma}_k^d) = \text{h}(\epsilon_k^x, \sigma_k^x) + \text{h}(\epsilon_k^y, \sigma_k^y) + \text{h}(\epsilon_k^z, \sigma_k^z) \qquad \text{h}(\epsilon, \sigma) = \begin{cases} \frac{1}{2}\epsilon^2(\sigma^2)^{-1} & \text{if } |\epsilon| < \delta, \\ \delta|\epsilon| - \frac{1}{2}\delta^2 & \text{otherwise} \end{cases} \tag{7}$$

We treat each axis separately because, as shown in later sections, state estimation errors along the $z$-axis are often quite different from the errors on the $xy$-plane. Our loss function resulted in 12% less error computing displacements in test experiments compared to [5].

## 5 State Estimation

This network outputs a relative position measurement and uncertainty which is suitable for incorporation into both filtering and smoothing frameworks, as described in the following sections.

### 5.1 Filtering Estimator

We incorporate the learned displacement measurement into Pronto, an open-source modular legged robot estimation framework [8]. As Pronto maintains a history of states, measurements and covariances, it can easily incorporate relative position measurements. The full system architecture is shown in Fig. 2: the IMU measurements from time $t_i$ to $t_j$ are accumulated and rotated to be gravity aligned at time $t_i$ using the orientation $\mathbf{R}_i$ from the filter's internal state. They are then passed to the network to estimate displacement and uncertainty. These are fused, using the covariance in Eq. 4, into the filter as a relative position measurement:

$$\mathbf{z}_j = \mathbf{p}_i + \hat{\mathbf{d}}_i \tag{8}$$

where position $\mathbf{p}_i$ at time $i$ is taken from the filter's history, while displacement is from the network.

We call our proposed filtering method EKF-IKD (EKF with inertial, kinematic, and learned displacement). For all training and experiments, we used a buffer size of $N = 400$ ($1\,\text{s}$ of $400\,\text{Hz}$ IMU data). Displacement measurements were estimated at $20\,\text{Hz}$, which are overlapping . We recognize that this overlap compromises independence of measurements but have not included analysis of this effect due to space limits.

### 5.2 Factor Graph Estimator

In this section, we describe a novel factor graph architecture that incorporates the learned inertial factor. In contrast to filters, factor graph methods re-optimize the full or partial history of states, given all the measurements connected to them, allowing for more accurate and smoother estimates.

Fig. 3 shows the proposed factor graph architecture. When a new camera keyframe is available, a new node (white circle) is created. Similarly to [17], we connect consecutive nodes $\boldsymbol{x}_k$ and $\boldsymbol{x}_{k+1}$ with preintegrated IMU and leg odometry factors (blue and orange factors), while visual features are modeled as individual landmarks (gray circles) and factors (yellow factors). In visually degraded environments, the visual landmarks might not be observable for a long period of time (e.g., from node $\boldsymbol{x}_3$ in the example), so it is important to have robust proprioceptive odometry to fall back on. To this end, we exploit the novel learned displacement factor to further constrain the robot motion

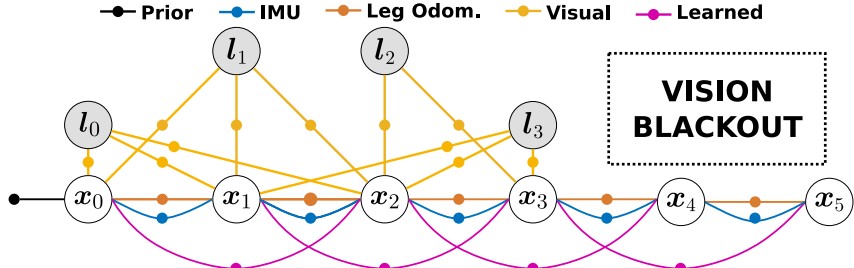

Figure 3: Proposed factor graph framework with IMU factor from [18], kinematics factor from [17] and our learned factor which specially helps in visual blackouts. The learned factor connections are dictated by the rates of the other sensors, as explained in Sec. 5.2.

(magenta line of Fig. 3). We call the factor graph implementation of our method FG-VIKD (factor graph with vision, IMU, kinematics and learned displacement).

The learned displacement measurement $\mathbf{d}_i$ estimates the position of the robot between the start and end times of the buffer. The learned factor residual $\mathbf{r_d}$ can then be computed using $\hat{\boldsymbol{\Sigma}}_i^d$ from Eq. 4:

$$\mathbf{r_d} = \left\| \mathbf{R}_i^\mathsf{T}(\mathbf{p}_j - \mathbf{p}_i) - \hat{\mathbf{d}}_i \right\|_{\hat{\boldsymbol{\Sigma}}_i^d}^2, \tag{9}$$

Fixed-lag optimization is done over the last 5 s of states each time a node is inserted to the graph. Since the IMU buffer is longer than the period between two keyframes, the nodes connected by the learned factor are not consecutive and instead skip several nodes. In the typical setup the camera creates nodes at 20 Hz and the IMU buffer length is 1 s meaning the learned factor would connect $\mathbf{x}_0$ to $\mathbf{x}_{20}$, $\mathbf{x}_1$ to $\mathbf{x}_{21}$ etc. (for simplicity, every two nodes are connected in Fig. 3). We also integrate IMU measurements between optimizations to provide a 400 Hz signal state estimate.

## 6   Lab Experiments

We performed several experiments with our ANYbotics ANYmal C100 robot as shown in Fig. 4. The robot was teleoperated over different terrains in the lab: flat ground, soft ground, a slippery surface and an obstacle. Ground truth poses were measured using a Vicon motion capture system.

**Flat Ground** First, we show performance on rigid, flat terrain. In these conditions, kinematic state estimate is very accurate, however over long trajectories there is upward drift (the Z dimension).

**Soft Ground** Deformable terrain poses a major challenge to legged robot state estimation as the specific moment of contact is ill-defined — the foot continues to move and compress the terrain after a contact is detected. This additional downward motion results in a hallucinated upward Z drift. Some realistic examples of deformable terrain include sand, mud or tall grass. To reproduce this effect we placed several layers of foam padding on the floor.

**Slipping** When a robot slips, such as on wet or loose terrain, certain kinematic assumptions are violated as the foot is in contact with the terrain but does not have zero velocity. This results in large spikes in odometry error. To create slip events in the lab, we placed a common office white board on the floor and covered it with lubricant (hand sanitizer).

**Obstacle Terrain** Blind climbing over an uneven obstacle course with ramps. This experiment was designed to show that our method can generalize to motions on non-planar surfaces.

Figure 4: Lab experiments. **Left:** Robot foot compressing into soft terrain. **Middle:** Slip event while walking on slippery terrain. **Right:** Robot climbing over an obstacle.

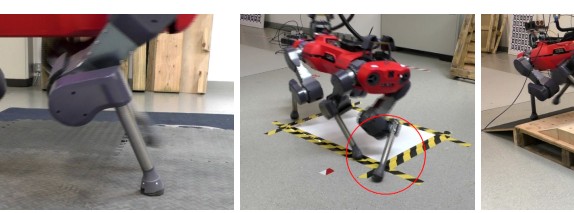

Table 1: Experimental Results for Filter-based methods.

| Data | Absolute Pose Error (APE) [m] | | | | 5 m Relative Pose Error (RPE) [m] | | | |
|------|------|------|--------|---------|------|------|--------|---------|
|      | TLIO | TSIF | EKF-IK | EKF-IKD | TLIO | TSIF | EKF-IK | EKF-IKD |
| FLAT | 1.88 | **0.14** | 0.42 | 0.20 | 0.58 | **0.09** | 0.19 | 0.16 |
| SLIP. | 0.28 | 0.21 | 0.38 | **0.20** | 0.35 | 0.13 | 0.20 | **0.12** |
| SOFT | 0.31 | 0.35 | 0.53 | **0.24** | 0.31 | **0.15** | 0.24 | 0.16 |
| OBST. | 0.37 | **0.19** | 0.39 | 0.27 | 0.44 | **0.12** | 0.22 | 0.22 |

## 6.1 Training

For the lab experiments, we collected ten walking sequences for each terrain totaling $2\,\text{h}16\,\text{min}$ in 40 sequences. Four sequences were selected as test experiments and the remaining were split 75:25 into training and validation datasets. We trained a single model from all the data and used this same model in all experiments. We used the Adam optimizer with an initial learning rate of $10^{-4}$ and a Huber delta of 1.35. Training lasted 200 epochs with the model minimizing validation error used in all lab experiments. Training took 6 hours on a laptop with an Nvidia Quadro P2000 GPU.

We used the Vicon system to collect training data. First, we remove the biases from the raw IMU signal and inject back random bias values. In order to estimate the true IMU biases, we create a factor graph where each node has a prior pose factor from a Vicon measurement. IMU factors connect the nodes allowing the biases to be estimated in batch optimization for all Vicon measurements.

## 6.2 Filtering Results

We present results comparing **filtering-based** proprioceptive estimator Pronto (which we hereafter refer to as EKF-IK) which is open-source and uses IMU and kinematics and our proposed method which incorporates IMU, kinematics and the learned displacement measurement EKF-IKD. We also show the trajectory provided by the default state estimator for the ANYmal robot which we call TSIF and uses IMU and kinematics as well as the learning-based estimator TLIO.

We used two metrics: absolute pose error (APE) and relative pose error (RPE) – as defined in [19]. APE measures the amount of drift over an entire experiment. RPE measures the drift every X m traveled – we used $5\,\text{m}$ for lab experiments and $10\,\text{m}$ for the mine experiment. In all experiments, adding the learned measurement improved the state estimation significantly. As shown in Table 1, in comparison with EKF-IK, APE was reduced by $46\,\%$ on average ($51\,\%$ for most challenging terrains: slipping and soft) and RPE was reduced by $22\,\%$ on average ($37\,\%$ on challenging terrains).

Our proposed method out performs TLIO for every experiment which is expected as we use kinematics. The TSIF estimator outperforms our method in situations where the terrain is rigid and stable but is on par when slipping or on soft terrain. We attribute this to the years of fine-tuning by the company on this estimator for this particular robot. As such, we make no claims of beating the state-of-the-art, however we point to the significant improvement gains to EKF-IK from the addition of the learned measurement.

**Flat Ground** The results for the filtering methods on flat ground are shown on the left of Fig. 5. There is a clear upward Z drift for TSIF and EKF-IK which is significantly reduced using the learned measurement in EKF-IKD .

**Soft Ground** The results for the filtering methods on soft ground are shown on the right of Fig. 5. The periods where the robot walked on the soft terrain are highlighted in gray. The upward Z drift

Table 2: Experimental Results for Factor-Graph methods.

| Data | Absolute Pose Error (APE) [m] | | 5 m Relative Pose Error (RPE) [m] | |
|------|------|--------|------|--------|
|      | FG-IK | FG-IKD | FG-IK | FG-IKD |
| FLAT | 0.36 | **0.22** | 0.25 | **0.24** |
| SLIP. | 0.23 | **0.20** | 0.18 | **0.17** |
| SOFT | 0.44 | **0.39** | 0.28 | **0.23** |
| OBST. | 0.23 | **0.21** | 0.18 | **0.18** |

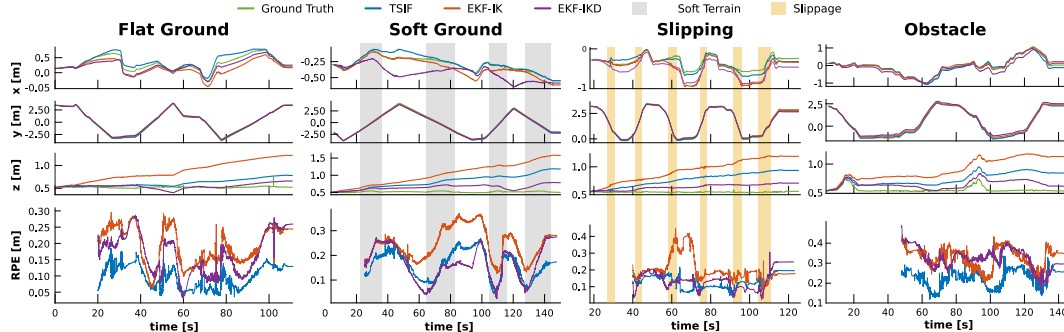

Figure 5: **Top:** Position in W estimate comparison for flat, soft, slippery, and obstacle grounds. Note the upward z drift for TSIF and EKF-IK on the soft (gray area) and slippery (yellow area) terrains. **Bottom:** $5\,\mathrm{m}$ Relative Pose Error (no data for first $5\,\mathrm{m}$ travelled).

for TSIF and EKF-IK is more pronounced but this is reduced in EKF-IKD. Along the $x$ axis the performance is slightly worse indicating scale error introduced by the learned measurement but overall error is greatly reduced as shown in the RPE plot.

**Slipping** As shown in Fig. 5, periods when the robot is slipping have high error. The addition of the learned measurement in EKF-IKD significantly reduces spikes in error, particularly at $60\,\mathrm{s}$. Our method does slightly better than TSIF in this experiment.

**Obstacle Terrain** This experiment demonstrates that our method can generalize to non-planar terrain. The upper right plot of Fig. 5 shows that EKF-IKD reduces Z drift while preserving the true changes in elevation as the robot climbs over the terrain. Initially, we found that if the training dataset did not include examples over elevated terrain, the network assumed constant planar elevation. The resulting state estimator then ignored elevation gain and loss from climbing on terrain.

### 6.3 Factor Graph Results

We present results comparing two **factor graph-based** proprioceptive estimators: FG-IKD which is our proposed factor graph implementation without vision data (simulating camera failure) and FG-IK which is the same factor graph without vision or the learned displacement. Since there is no vision in these cases, nodes are created at a $40\,\mathrm{Hz}$ fixed frequency in sync with the IMU.

We demonstrate that by adding the learned factor to this factor graph without any vision, APE was reduced by $18\,\%$ and RPE by $7\,\%$, as shown in Table 2. The error in these factor graph methods is higher than the filtering methods because the robot state is intentionally estimated at a lower rate to allow visual data to be incorporated. Additionally, without landmarks, FG-IK cannot take advantage of the graph structure. Nevertheless, FG-IKD shows an improvement in proprioceptive estimation implying that if a robot's vision system were to fail, our method would reduce drift.

## 7  Field Experiment: Underground Mine

While the learned displacement measurement can improve proprioceptive state estimation, most deployments of robots use vision systems. However, performance of these systems will degrade in challenging environments such as underground mines. In these cases, the learned displacement measurement can support the vision-based state estimation system. To demonstrate this, we performed a field experiment in which we teleoperated our ANYmal robot through a decommissioned limestone

Table 3: Experimental results for the Mine Experiment.

| Data | Absolute Pose Error (APE) [m] | | | | 10 m Relative Pose Error (RPE)[m] | | | |
|------|------|---------|--------|----------|------|---------|--------|----------|
| | TLIO | EKF-IKD | FG-VIK | FG-VIKD | TLIO | EKF-IKD | FG-VIK | FG-VIKD |
| MINE | 3.37 | 5.43 | 4.63 | **4.09** | 0.66 | 0.49 | 0.45 | **0.35** |

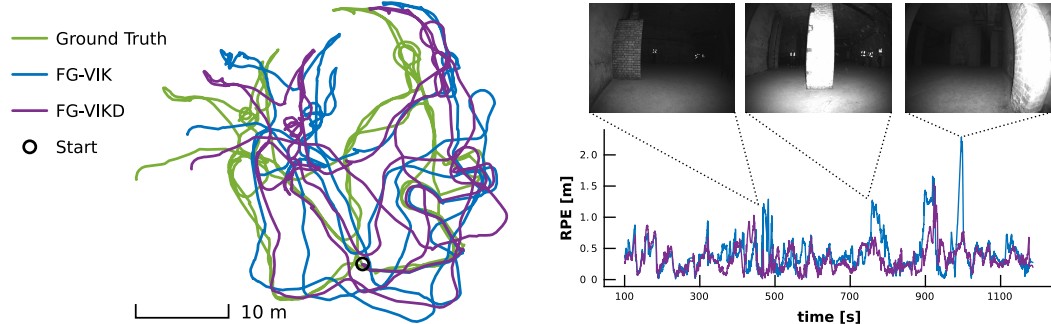

Figure 6: Results from the mine experiment. **Left:** Top-down perspective of FG-VIK vision-based factor graph and our method FG-VIKD in W. **Right:** 10 m RPE for both trajectories. We show images from three times where the error spikes in the baseline FG-VIK due to poor visibility.

mine with large rooms and narrow passageways as shown in Fig. 1. The floor is mostly hard with flat and sloped sections, but thick layers of dust and loose debris create slippery areas.

In the mine the controller from [20] was used instead of the default controller. Thus a new model was trained from lab data as described in Section 6.1, totaling 1 h 23 min. Training data for the mine was collected entirely in the lab and the resulting model was successfully applied to the mine.

### 7.1 Mine Experiment Results

We compared a classical factor graph-based state estimator VILENS [10]; which uses vision, IMU and kinematics (and hereafter we refer to as FG-VIK); to our method FG-VIKD which uses the same sensors as well as the learned displacement factor. In the mine visual sensing is severely affected, forcing FG-VIK to rely on its proprioceptive sensing and, in these situations, the learned factor can help keep error low. Our experiment lasted 20 minutes and over 472 m of walking.

Ground truth was estimated using the robot's lidar to localize in a prior, high quality, point cloud map. Table 3 shows that RPE was reduced by 22 % by adding the learned factor. We also show TLIO and EKF-IKD as baselines although they do not use cameras. The trajectory and error over time are shown in Fig. 6 along with images from the front facing camera at times where the drift rate spiked. These spikes coincide with periods of poor illumination and visual tracking failure in the dark environment. The robot lights reflected off nearby surfaces triggering the camera to reduce its exposure making the image darker. The performance improvement would likely be reduced in radically different terrain such as sand or snow; however, this experiment demonstrates generalization from the artificial lab terrains to real-world environments.

## 8  Conclusion

In this paper, we introduced a learned displacement measurement derived from inertial sensing which can greatly reduce state estimation drift for legged robots in challenging situations. We showed how this measurement can be used in both filtering and factor graph frameworks, and that the greatest benefits are when the robot slips or the terrain is deformable. Extensive experiments with the ANYmal quadruped robot showed a reduction of 37 % in relative pose error for filter-based state estimation. We have also shown how the learned displaced measurement can reduce error in a vision driven factor graph framework by 22 % in a underground mine. A video demonstrating these experiments is provided as supplementary material.

For future work, inclusion of kinematic data into the network will be explored. Directly using the joint states as input to the network would likely require a much more complex architecture that might not generalize to different robot models. To this end, we seek to incorporate derivative inputs from the robot model, such as ground reaction forces or foot velocities.

**Acknowledgments**

This research has been conducted as part of the ANYbotics research community. It was part funded by the EU H2020 Project THING (Grant ID 780883) and a Royal Society University Research Fellowship (Fallon).

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
