# OpenReview forum: "Learning Inertial Odometry for Dynamic Legged Robot State Estimation"
_robot-learning.org/CoRL/2021/Conference — CoRL2021 Poster_

### Official Review · Reviewer_twnh · 2021-07-21

**Originality:** Fair
**Technical Quality:** Fair
**Clarity Of Presentation:** Good
**Impact:** 2

**Recommendation:**

Weak Reject: I recommend rejecting the paper, but will not argue for my recommendation if the majority of other reviewers have a different opinion.

**Summary:**

This paper proposes a learned state estimator that can be fused with kinematics and IMU measurements as well as factor graphs to estimate robot state over long experiments, without significant drift. Authors present experiments in a mine where the camera image clearly shows a lot of noise that would make estimation hard using visual features. In such cases, the learned estimator can augment visual estimation reduce the robot state estimation drift.

**Issues:**

Main issues:
1. No learning-based baselines. Since approach trains a model on each terrain separately, it gives the approach significant advantage over baselines, making comparisons unfair.
2. only a 3D pose estimate, not 6-dof estimates of state presented in the paper.

**Reviewer Expertise:**

Very good: Comprehensive knowledge of the area

**Strengths And Weaknesses:**

Strengths:
1. The paper is well-written and the use of learned models especially in the presence of large visual noise is well motivated.
2. The robot experiments are well-conducted and run over long episodes, increasing the confidence in the performance of the estimator.
3. The paper presents ways of using the proposed approach with both extended kalman filters that take IMU and IK data, as well as factor graphs that can take visual data as input. This makes the method versatile and easy to adapt to new problems.

Weaknesses:
1. The proposed approach requires re-learning of the estimator on each terrain. This is a major shortcoming of the approach as the major advantage of a learned estimator would be generalization to unseen terrains, where the parameters of baseline filters cannot be hand-designed. There is no discussion of this in the paper.
2. The proposed approach only estimates the x,y,z position of the robot, not the orientation. This is also a major shortcoming which is not discussed. I would suspect that in situations such as slipping, the orientation estimate goes inaccurate, but no data is provided.
3. While the paper compares their approach against Pronto which uses IMU and IK information, they do not compare against any learning-based baselines. Since the proposed approach learns an estimator on the ground profile it is tested on, but the baselines do not, the comparison is not fair. Some learning based baselines that might make experiments more compelling are [a], [b], [c]. A simple baseline would even be to learn the dynamics model used in the EKF.
[a] Liu, Wenxin, et al. "TLIO: Tight learned inertial odometry." IEEE Robotics and Automation Letters 5.4 (2020): 5653-5660.
[b] Kim, Youngji, et al. "Unsupervised Balanced Covariance Learning for Visual-Inertial Sensor Fusion." IEEE Robotics and Automation Letters 6.2 (2021): 819-826.
[c] Teng, Sangli, Mark Wilfried Mueller, and Koushil Sreenath. "Legged Robot State Estimation in Slippery Environments Using Invariant Extended Kalman Filter with Velocity Update." arXiv preprint arXiv:2104.04238 (2021).

Detailed comments:
1. The introduction should clarify that the proposed estimator only applies to 3D positions and not orientations.
2. Eq 1 and Eq 2 are mixing discrete and continuous time conventions, and variables k, t
3. Why do authors use MSE to start and switch to GML loss? What was the motivation behind this choice?
4. Is each dimension considered independent is Eq 7? Is this a reasonable assumption, especially in situations like slipping? Can authors motivate this choice?
5. Why do authors ignore covariance estimates when predicting state in Eq 8? Could covariances be useful in fusing the estimates from filter and NN better than simply adding the displacements?
6. Section 5.2 is not very well described. What happens between visual events? Is the past estimate of state held constant or is the EKF run on proprioceptive data?
7. In Section 6.1, is the data of 2h16m per terrain or across all terrains? Is the same model fit to all terrains or one per terrain?
8. What is the frame of reference for experiments in Figure 5, 6, etc. Is y-movement forward?
9. The main advantage of the proposed approach is in keeping the z-direction drift. However, in the experiments the z remains mostly fixed (except on ramp, where TSIF outperforms the proposed approach). The NN could just be learning to keep the z fixed. Moreover, the performance does not look better in x and y directions.  More discussion on poor performance in x on soft ground would be useful.
10. How different is the mine from lab experiments? Though the video is impressive, overall the generalization to the mine terrain from lab is unconvincing. Why do authors use a vision-based approach here? How does EKF-IKD perform in this setting? This section needs to be motivated better and the choice of controller, approach, etc be described in more details.


**Summary Of Recommendation:**

The paper presents an incremental approach to estimation which involves a trained network predicts displacements in 3D positions based on IMU history. Similar ideas are present in literature on 6-dof pose estimates, like [1]. The paper could benefit from learning based baselines, recommended in the main review.
[1]  Silva do Monte Lima, João Paulo, Hideaki Uchiyama, and Rin-ichiro Taniguchi. "End-to-end learning framework for imu-based 6-dof odometry." Sensors 19.17 (2019): 3777.

---

> ### Author Response · Authors · 2021-08-30
> **Response to Reviewer twnh (1/2)**
>
> We thank the reviewer for their comments and suggestions, we have made several changes to the manuscript to make it more clear.
>
> ### A summary of changes to the paper is provided here:
>
> - Added discussion of generalization to different terrain.
> - Added learning-based baseline.
> - Clarified some potential misunderstandings, for example, we do estimate 6 DoF pose and do not need to retrain for every terrain.
>
> ### More detailed responses are provided below:
>
> **1. The proposed approach requires re-learning of the estimator on each terrain. This is a major shortcoming of the approach as the major advantage of a learned estimator would be generalization to unseen terrains, where the parameters of baseline filters cannot be hand-designed. There is no discussion of this in the paper.**
>
> We thank the reviewer for their comments and suggestions. We would like to clarify that re-learning is not necessary for each terrain.
>
> Firstly, equal amounts of data was collected for each terrain type and compiled into a single dataset from which a single model was trained. The same model was for all terrain in the lab experiments. We do not train separate models for each terrain type. We believe this may have been a misunderstanding and have modified the manuscript in Section 6.1 to be clearer. This shows the network can learn across a range of different terrains. Our method does not need the exact terrain type to be known in advance.
>
> Secondly, for the mine experiment, a similar dataset was collected of the terrains in the lab but with the robot using a different controller. This model was then deployed in the completely new environment of the mine. The ground in the mine is mostly flat and hard but also slippery at certain points due to dirt and debris. We believe this demonstrates that training on a simple range of terrain stiffness and slipperiness can sufficiently cover most terrain the robot might encounter in the real world and therefore does not require re-training for each terrain or environment. To make this point more clear in the manuscript we have updated Section 7.
>
> We do acknowledge that if tested on significantly different terrain compared to the training set (for example sand) then performance would suffer. To address this we have also provided more discussion of out-of-terrain limitations in Section 7.1.
>
> **2. The proposed approach only estimates the x,y,z position of the robot, not the orientation. This is also a major shortcoming which is not discussed. I would suspect that in situations such as slipping, the orientation estimate goes inaccurate, but no data is provided.**
>
> We thank the reviewer for their comments. We believe there may be some misunderstanding. The proposed method does in fact estimate the full 6DoF pose of the robot. While the learning component estimates position displacement, the fusion of other sensors with EKF and Factor Graph backends provide the full robot state as described in Section 3.
>
> **3. While the paper compares their approach against Pronto which uses IMU and IK information, they do not compare against any learning-based baselines. Since the proposed approach learns an estimator on the ground profile it is tested on, but the baselines do not, the comparison is not fair. Some learning based baselines that might make experiments more compelling are [a], [b], [c]. A simple baseline would even be to learn the dynamics model used in the EKF.**
>
> We thank the reviewer for their suggestion and have added [a] (TLIO) as a baseline for all lab experiments.
>
> **4. The introduction should clarify that the proposed estimator only applies to 3D positions and not orientations.**
>
> We believe there has been a misunderstanding as our method does estimate the full 6 DoF pose of the robot. The learned component produces a displacement but the full system fuses this with classically methods and computes orientations as well.
>
> **5. Eq 1 and Eq 2 are mixing discrete and continuous time conventions, and variables k, t**
>
> We thank the reviewer for pointing this out. In our paper we followed the same convention as Forster2017, but due to the limited space the relationship between k and t was not clear. We have modified the text to improve it.
>
> **6. Why do authors use MSE to start and switch to GML loss? What was the motivation behind this choice?**
>
> We use the GML-Huber loss to train the network to produce uncertainty values on the measurements. However, we find that if starting directly with this loss the network does not converge and so we initialize training with the MSE loss. We have changed the wording in Section 4.1 slightly to make this more clear.

---

> > ### Author Response · Authors · 2021-08-30
> > **Response to Reviewer twnh (2/2)**
> >
> > **7. Is each dimension considered independent is Eq 7? Is this a reasonable assumption, especially in situations like slipping? Can authors motivate this choice?**
> >
> > This is a very relevant question. Yes, we do treat each axis separately when computing the Huber loss. This is based on the observation that error in the Z direction is often quite different from error in the xy plane particularly in dynamic situations. This allows for  a bad estimate in one axis to not invalidate a good estimate in others. We have added a line to this section to discuss this.
> >
> > **8. Why do authors ignore covariance estimates when predicting state in Eq 8? Could covariances be useful in fusing the estimates from filter and NN better than simply adding the displacements?**
> >
> > We do use the covariance estimate for fusion. Eq 8 is the measurement update which, in addition to the covariance, is fused into the EKF and Factor Graph. We have updated the text to make this more clear.
> >
> > **9. Section 5.2 is not very well described. What happens between visual events? Is the past estimate of state held constant or is the EKF run on proprioceptive data?**
> >
> > Thank you for this comment, other reviewers have also felt our explanation of the factor graph could be improved and as such we have edited Section 5.2 to provide more explanation. This section describes the Factor Graph implementation of our method, not the EKF. In a factor graph, the entire history of states is recorded. In our implementation we optimize over a fixed window of the last couple of states. To have a high rate state estimator, between nodes we simply integrate the IMU data and therefore have a 20 Hz (camera rate) optimized state and 400 Hz (IMU rate) propagated state from the last optimized state. We have added these details to the paper.
> >
> > **10. In Section 6.1, is the data of 2h16m per terrain or across all terrains? Is the same model fit to all terrains or one per terrain?**
> >
> > Thank you for asking this question. We realize the section was not very clear and have updated Section 6.1. The dataset of 2h16m comprised 40 sequences, 10 for each terrain type. A single model was trained on all data and used in all lab experiments.
> >
> > **11. What is the frame of reference for experiments in Figure 5, 6, etc. Is y-movement forward?**
> >
> > Position results are shown in the fixed world frame W, as such no axis can be said to be the “forward” direction. We have updated the figure captions to indicate the frame.
> >
> > **12. The NN could just be learning to keep the z fixed. Moreover, the performance does not look better in x and y directions. More discussion on poor performance in x on soft ground would be useful.**
> >
> > We thank the reviewer for their comments. As we explained in the manuscript the purpose of the obstacle experiment was to show how the upward Z drift is reduced but not to the point of forcing the z position to be constantly zero. If the NN were learning a zero Z displacement, then the peaks of the Z trajectory would be significantly more flat, particularly the first peak.
> >
> > We agree with the reviewer that the performance in X  appears to be worse in EKF-IKD. Looking only at the position trajectories can be somewhat misleading as the scales are different to make the separate trajectories more visible. This is why we show RPE plots below the trajectories to show the overall error. We have added more discussion of this to the paper.
> >
> > **13. How different is the mine from lab experiments? Though the video is impressive, overall the generalization to the mine terrain from lab is unconvincing. Why do authors use a vision-based approach here? How does EKF-IKD perform in this setting? This section needs to be motivated better and the choice of controller, approach, etc be described in more details.**
> >
> > Thank you for the comment. The mine is quite different from the lab, we are unsure what is particularly unconvincing to the reviewer about the generalization from lab to mine. The ground is mostly flat and hard but with slopes and loose debris which causes slipping. There are also wet areas creating a slippery surface. The mine is very similar to the one used in the DARPA SubT Challenge Tunnel Circuit (https://www.youtube.com/watch?v=LAziR-R-07c) except there are no areas with deep mud through which the robot would not have been able to walk anyway.
> >
> > We have included EKF-IKD in Table 3.
> >
> > Regarding motivation we agree with the reviewer that better motivation for this experiment is needed and have updated section 7 in the manuscript. We explain that while in previous sections we sought to improve proprioceptive methods, in Section 7 we demonstrate improvements to vision systems in visually degraded environments. Additionally, we provide more information on the experimental setup and, as the review suggests, we have added EKF-IKD to Table 3.

---

> > > ### Comment · Reviewer_twnh · 2021-09-02
> > > **thanks**
> > >
> > > Thank you for your very detailed and well though-out response. I appreciate the clarifications, updates to the paper and the added comparison experiment. I am willing to change my rating to weak accept based on author's response.

---

### Official Review · Reviewer_VSob · 2021-07-23

**Originality:** Good
**Technical Quality:** Good
**Clarity Of Presentation:** Very Good
**Impact:** 3

**Recommendation:**

Weak Accept: I recommend accepting the paper, but will not argue for my recommendation if the majority of other reviewers have a different opinion.

**Summary:**

The paper introduces a neural network for state estimation of a legged robot. The NN outputs positional displacements and their corresponding uncertainty based on raw IMU data. These displacements are then integrated into prior methods for state estimation with multiple sensors to output a more robust pose (with reduced sensor drift). Experiments were conducted for challenging terrains (like slippery or soft surfaces) and in a real-world mine environment.

**Issues:**

- We conclude from the experimental results that the proposed method performs worse than related work: For example, in comparison between the proposed EKF-IKD and the default TSIF state estimator, the proposed method performs 40% worse on average (RPE) in the lab experiments (Table 1). The abstract states a reduction by 37% in comparison to a traditional kinematic-inertial estimator, but doesn’t that include TSIF? Apart from the drift of the z-axis, the TSIF also shows qualitatively better results in Fig. 5 and 6. Why are the best results of the factor graph-based methods highlighted, although they are just different methods for solving the same task? We strongly suggest adding a baseline for the mine experiment.
- While it is not inherently bad that the results are not better than related work, we think that the paper should acknowledge and might discuss reasons for that.
- Again, we think that the discussion does not emphasize the obvious disadvantages of the learned approach. We suggest adding experiments for out-of-distribution terrains (like surfaces with different softness) to show the robustness of the proposed work. In particular, the robustness is a key motivation of your work (and arguably very important for real-world usage of your state estimator). In robot learning, we find it very important to get a feeling about the limitations of the approach. Why doesn’t the learned approach fix the drift in the z-direction completly?
- Why doesn’t the NN get the joint states q as input? We think that the NN could detect (and then correct) challenging situations in particular based on a mismatch between IMU and joint data.
- We suggest clearing up Section 3. From our perspective, there is no real “problem statement” or task in there. What should be minimized or calculated? The biases b are not used furthermore (as they are learned). Between which points of time are the position increments d_i defined?
- Fig. 7 (left side) should be cleared up, e.g. by combining all three trajectories into one figure. As no start point is mentioned, it might make sense to shift and rotate the trajectories by a constant offset first.
- The title of the tables should be above.
- To make room for additional discussion: Fig. 5 and 6 could be made smaller and combined into one, and Section 2 and 5 could be shortened.


**Reviewer Expertise:**

Fair: Some knowledge of the area

**Strengths And Weaknesses:**

Integrating a learned approach for state estimation is a powerful idea that might improve the handling of challenging situations. However, the experimental results show that the proposed IKD methods perform worse than the default TSIF method. Nonetheless, the learned method improves the corresponding approach without learning, however only with very similar training and test data distribution. We think that the paper does not put enough emphasis on the limitations of learning: a possible distribution mismatch and the (probably constrained) generalization capability compared to a model-based approach.

**Summary Of Recommendation:**

The paper has an interesting idea, however the evaluation and discussion should be extended to support the claims (e.g. “our approach greatly reduces the drift of ...” in the abstract).

---

> ### Author Response · Authors · 2021-08-30
> **Response to Reviewer VSob (1/2)**
>
> We would like to thank the reviewer for their careful consideration of the manuscript. We have made several changes to the text in order to provide more discussion of results and analysis of the limitations of our method.
>
> ### A summary of changes to the paper is provided here:
>
> - Added additional discussion of performance of our method.
> - Added additional discussion of out-of-distribution testing
> - Added Discussion of kinematics as input
> - Changes to Figures
> - Added learning-based baseline
>
> ### More detailed responses are provided below:
>
> **1. The abstract states a reduction by 37% in comparison to a traditional kinematic-inertial estimator, but doesn’t that include TSIF? Apart from the drift of the z-axis, the TSIF also shows qualitatively better results in Fig. 5 and 6. Why are the best results of the factor graph-based methods highlighted, although they are just different methods for solving the same task? We strongly suggest adding a baseline for the mine experiment.**
>
> We would like to thank the reviewer for their careful consideration of the manuscript. We have made several changes to the text in order to provide more discussion of results and analysis of the limitations of our method.
>
> We present the trajectory from TSIF which is initially based on the work of Bloesch et al., but has since been significantly fine-tuned for the ANYmal robot by the company and is closed-source. However, we felt we could not simply exclude the results from TSIF. This is in contrast to Pronto which is designed to work with many different kinds of legged robots and is open source, as such we could only adapt Pronto to use the learned displacement measurement.
>
> Regarding the 37% reduction we agree the wording should be changed however we feel it is significant that there is such an improvement to Pronto in adding the learned measurement. We believe the takeaway should be that adding the learned displacement can improve state estimation, not necessarily that the full proposed methods are better than algorithms which are carefully tuned to a specific robot platform. If it were possible to add to TSIF we would of course do that as well. We have changed the wording in the abstract and in Section 6.2 to make this point more clear.
>
> Regarding Fig. 5 and 6, looking only at the position trajectories can be somewhat misleading as the scales are different to make the separate trajectories more visible. In practice, 10 cm error in the x-direction might be acceptable if error in the z direction is reduced by 1 m. This is why we show RPE plots below the trajectories. Again, we don’t claim to be more accurate than the TSIF estimator but simply aim to show that in dynamic or challenging situations the learned displacement measurement can help reduce total odometry drift.
>
> We agree the Table 1 was confusing and have removed the factor graph methods to a separate table. We believe it is important to consider filtering and optimized methods separately because the back end optimization is different.
>
> We have added a baseline to the mine experiment in Table 3.
>
> **2. While it is not inherently bad that the results are not better than related work, we think that the paper should acknowledge and might discuss reasons for that.**
>
> We thank you for this comment and agree that more analysis of our results should be provided. We have rewritten Sections 6.2 to this effect.
>
> **3. We suggest adding experiments for out-of-distribution terrains (like surfaces with different softness) to show the robustness of the proposed work. In particular, the robustness is a key motivation of your work (and arguably very important for real-world usage of your state estimator). In robot learning, we find it very important to get a feeling about the limitations of the approach. Why doesn’t the learned approach fix the drift in the z-direction completly?**
>
> Thank you for the comments. We agree that the manuscript could do more to address the limitations of our method and have added more discussion in Sections 6.2 and 6.3.
>
> Regarding out-of-distribution experiments we consider the mine experiment to be an example of an out-of-distribution experiment as the model was trained entirely with artificial terrain in the lab. The terrain in the mine is varied in terms of slopes and slipperiness and we have updated the text to emphasize this point. Of course, a significantly different terrain to any in the training set would likely affect the results and we have added a discussion of this to Section 7.1.
>
> Regarding why the drift isn’t completely eliminated, no odometry system is completely drift free. Additionally, the back-end sensor fusion of either the EKF or Factor Graph must still include the drift introduced by the kinematic information. This error can only be reduced by using a more accurate sensing modality with higher confidence.

---

> > ### Author Response · Authors · 2021-08-30
> > **Response to Reviewer VSob (2/2)**
> >
> > **4. Why doesn’t the NN get the joint states q as input? We think that the NN could detect (and then correct) challenging situations in particular based on a mismatch between IMU and joint data.**
> >
> > We thank the reviewer for their useful suggestion. We had considered using the joint states as additional input to the network but preferred to start with a simpler system that is easier to characterize. We believe adding q directly as input would force the network to learn the forward kinematics of the specific legged robot which would both require a much larger network and make the method less flexible. Instead, we have plans for future work to incorporate preprocessed data from the joints such as end effector velocities or ground reaction forces. We have added discussion of this to the conclusion.
> >
> > **5. We suggest clearing up Section 3. From our perspective, there is no real “problem statement” or task in there. What should be minimized or calculated?  The biases b are not used furthermore (as they are learned). Between which points of time are the position increments d_i defined?**
> >
> > Thank you for your comments, we would disagree with the assertion that “there is no problem statement”. As we write in the text the objective is to estimate the robot state x_k which includes the full 6 DoF pose of the robot, velocity and biases.
> >
> > The network learns a motion prior of the robot which does likely involve learning a representation of the biases, however it is not right to say they are not used further. The EKF and factor graph back-ends will fuse the various sensing modalities and estimate the IMU biases.
> >
> > The network takes a buffer of IMU data and produces a displacement between the first and last time of the buffer. We have added a line to Section 4 to make this more clear.
> >
> > **6. Fig. 7 (left side) should be cleared up, e.g. by combining all three trajectories into one figure. As no start point is mentioned, it might make sense to shift and rotate the trajectories by a constant offset first.**
> >
> > We have combined the three trajectories into one figure and indicated the start point in the legend.
> >
> > **7. The title of the tables should be above.**
> >
> > Thank you, this has been updated in the manuscript.
> >
> > **8. To make room for additional discussion: Fig. 5 and 6 could be made smaller and combined into one, and Section 2 and 5 could be shortened.**
> >
> > As suggested by the reviewer, we have combined Fig 5 and 6 and we have shortened sections 2 and 5.

---

> > > ### Comment · Reviewer_VSob · 2021-09-03
> > > **Response**
> > >
> > > Thanks for your detailed answer! I think your revision has addressed (most) of my issues and has improved the paper. I'll change my overall recommendation to weak accept.
> > >
> > > Regarding your sentence: "We believe the takeaway should be that adding the learned displacement can improve state estimation, not necessarily that the full proposed methods are better than algorithms which are carefully tuned to a specific robot platform." Please keep in mind that you use a learned approach without evaluating the generalization to other robots, so I would call your approach "carefully tuned to a specific robot platform" as well.

---

### Official Review · Reviewer_Kasg · 2021-07-23

**Originality:** Fair
**Technical Quality:** Fair
**Clarity Of Presentation:** Good
**Impact:** 3

**Recommendation:**

Weak Accept: I recommend accepting the paper, but will not argue for my recommendation if the majority of other reviewers have a different opinion.

**Summary:**

The paper proposes a method to estimate the position and orientation of a legged robot by combining the IMU data, the robot kinematics and visual observations. A focus of the paper is on a drift-free inertial odometry that uses a recurrent neural network. Another is on the application of the factor graph in order to exploit observations of landmarks that might be intermittently available. The method is effective particularly in environments where firm contacts between the robot feet and the ground are not available and the vision guidance is hardly expected due to bad optical conditions.


**Issues:**

To clearly explain the technique around the factor graph.


**Reviewer Expertise:**

Very good: Comprehensive knowledge of the area

**Strengths And Weaknesses:**

It seems reasonable to apply the neural network to the inertial odometry that might be able to lower the effect of sensor drift as well as the accumulation of numerical errors, so that a more robust estimation than a naive integral of the acceleration is expected with it. On the other hand, it is suspected that the method reduces the accuracy if the sufficient number of training data under sufficient variety of situations are not provided. The results in Table 1 pretty matches this prediction, and thus, is not surprising for the reviewer.

The technique around the factor graph is not clearly explained. Although the reviewer understands that the original technique by the authors is the use of learned transitions of the robot states (magenta lines in Figure 3), he/she does not understand why it connects every other nodes. It would be strongly suggested that the authors carefully explain how the graph is constructed and how several hypotheses of transitions between nodes including that based on the learned process are combined.

Regarding the results in Figure 7 (with a view of the supplement movie), the reviewer can hardly justify the claim that FG-VIKD is better than FG-VIK since the closeness of the latest positions of each trajectory to the ground truth seems to be frequently switched.


**Summary Of Recommendation:**

I recommend accepting the paper, but will not argue for my recommendation if the majority of other reviewers have a different opinion.

---

> ### Author Response · Authors · 2021-08-30
> **Response to Reviewer Kasg**
>
> We would like to thank the reviewer for comments and suggestions. We have made several changes to the text in order to make the manuscript more clear.
>
> ### A summary of changes to the paper is provided here:
> - Additional explanation of Factor Graph
>
> ### More detailed responses are provided below:
>
> **1. It seems reasonable to apply the neural network to the inertial odometry that might be able to lower the effect of sensor drift as well as the accumulation of numerical errors, so that a more robust estimation than a naive integral of the acceleration is expected with it. On the other hand, it is suspected that the method reduces the accuracy if the sufficient number of training data under sufficient variety of situations are not provided. The results in Table 1 pretty matches this prediction, and thus, is not surprising for the reviewer.**
>
> We agree with the reviewer that with less training data the performance would be worse and likely with more data we could get a boost in performance. In fact, in our preliminary experiments we found that if the robot was trained on slow walking data then tested on faster trotting data, the model did not generalize. For example, if trained on data with max speed 0.1 m/s then tested on 0.2m/s the estimated trajectory was almost exactly half the length of the true trajectory.
>
>  We intend to show in this paper that for the amount of data collection we have done a meaningful improvement to state estimation can be gained. We do believe that in our manuscript there has been some unclarity in the training process and have updated Section 6.1 to improve clarity.
>
> **2. The technique around the factor graph is not clearly explained.**
>
> We thank the reviewer for pointing out the lack of clarity. The magenta line connections are not exactly representative of how the factor graph is really constructed. We have clarified this by updating the caption of Fig. 3 and updating the text in Section 5.4
>
> **3. Regarding the results in Figure 7 (with a view of the supplement movie), the reviewer can hardly justify the claim that FG-VIKD is better than FG-VIK since the closeness of the latest positions of each trajectory to the ground truth seems to be frequently switched.**
>
> We thank the reviewer for their comment. We would like to point out the additional quantitative analysis of the trajectories particularly of the 10m RPE and APE in Table 3 which demonstrate that FG-VIKD is indeed better than FG-VIK.

---

> > ### Comment · Reviewer_Kasg · 2021-09-03
> > **Revision not found.**
> >
> > The reviewer is afraid that the revised manuscript is not found. Hence, he/she does not change the evaluation.

---

> > > ### Author Response · Authors · 2021-09-03
> > > **RE: Revision not found**
> > >
> > > The revised version is the latest uploaded version available by clicking the PDF button above. The link is also available here:
> > > https://openreview.net/pdf?id=a5ZiDzL0enJ
> > >
> > > Clicking on "Show Revisions" will also allow the reviewer to compare the differences between original and revised versions.

---

> > > > ### Comment · Reviewer_Kasg · 2021-09-03
> > > > **Sorry for misunderstanding**
> > > >
> > > > The reviewer noticed that the uploaded version is not the same with the first submission as pointed out by the authors, and is sorry about it.

---

### Official Review · Reviewer_je1b · 2021-07-25

**Originality:** Fair
**Technical Quality:** Very Good
**Clarity Of Presentation:** Very Good
**Impact:** 3

**Recommendation:**

Weak Accept: I recommend accepting the paper, but will not argue for my recommendation if the majority of other reviewers have a different opinion.

**Summary:**

This paper proposes a data-driven estimator for the odometry estimation of a legged robot from IMU data. In particular, the authors propose to learn a 1-D CNN to predict the displacement vector from a history of IMU states. They demonstrate the benefit of their learning-based odometry estimation by combining it with odometry from leg kinematics in an EKF estimator and in a factor graph estimator with a visual blackout. The proposed method improves relative pose error by 37 % in challenging scenarios when compared to traditional kinematic-inertial estimators.


**Issues:**

One modification that would help in understanding the different components of the proposed method would be to include ablations that plot the pose error as the history length of IMU changes and also see how the performance changes with the loss function used in [5] instead of the Huber loss function described in Section 4.1 (eqn 7). Although this is not critical for the acceptance of the paper, this should definitely be included in the final version of the paper.



**Reviewer Expertise:**

Fair: Some knowledge of the area

**Strengths And Weaknesses:**

Comments:
1. The proposed method is quite simple and elegant and seems to substantially improve the results compared to the state of the art. Using a learned prediction model to predict the displacement vector from buffered IMU data is a good place to leverage data-driven priors
2. Section 6.2, 6.3 and 7.1 do a great job of extensively evaluating the proposed method with comparisons to all the relevant non-vision baselines. in particular, the real-world evaluation is quite convincing.
3. Although I understand that the scope of the paper is limited to improving odometry with missing vision/lidar input, it is unclear to me how practical is a setup with completely missing vision. For example, it would be interesting to see the effective improvements when a learning-based Visual Odometry method [A] is also adding estimates to the factor graph from very noisy lidar/perception inputs. I suspect the gains would not be as much in that case.
4. It would be interesting to see ablations that plot the pose error as the history length of IMU changes, and also see how the performance changes with the loss function used in [5] instead of the huber loss function described in Section 4.1 (eqn 7).

[A] Wang, Sen, et al. "Deepvo: Towards end-to-end visual odometry with deep recurrent convolutional neural networks." 2017 IEEE International Conference on Robotics and Automation (ICRA). IEEE, 2017.


**Summary Of Recommendation:**

Although, I have some general concerns related to the specific setup of completely missing vision sensors, and the effective benefits of the proposed method with learning-based visual odometry which can potentially handle very noise perceptual inputs, the proposed method will still contribute to improved odometry since the contributions of the paper are complementary to my concern. That setup might reduce the performance gains of the proposed method, but my concern is more to understand what problems practically occur in the field of odometry estimation. I think the paper is quite strong in empirical demonstration of its results. I am inclined to accept the paper.

---

> ### Author Response · Authors · 2021-08-30
> **Response to Reviewer je1b**
>
> We would like to thank the reviewer for comments and suggestions. We have since made several changes to the manuscript.
>
> ### A summary of changes to the paper is provided here:
>
> - Added information on performance changes from using Huber loss function.
>
> ### More detailed responses are provided below:
>
> **1. Although I understand that the scope of the paper is limited to improving odometry with missing vision/lidar input, it is unclear to me how practical is a setup with completely missing vision It would be interesting to see the effective improvements when a learning-based Visual Odometry method [A] is also adding estimates to the factor graph from very noisy lidar/perception inputs.**
>
> We would like to thank the reviewer for comments and suggestions. We have since made several changes to the manuscript.
> Regarding the practicality of total vision system loss, legged robot locomotion and odometry is defined by performance in the worst case scenarios. Many labs e.g. Boston Dynamics, Agility developed feedback control with no extraception before adding vision/lidar mapping [1].
>
> [1] “Blind Bipedal Stair Traversal via Sim-to-Real Reinforcement Learning” Siekmann et al.
>
> Regarding a learning based baseline, other reviewers have also suggested we include a learning-based baseline and to that end we have updated the manuscript to include results from TLIO.
>
> **2. It would be interesting to see ablations that plot the pose error as the history length of IMU changes, and also see how the performance changes with the loss function used in TLIO instead of the huber loss function described in Section 4.1 (eqn 7).**
>
> We thank the reviewer for this suggestion. Our network builds on the one from TLIO and in that paper they provide an ablation study of different window lengths, as such we omit one in this paper due to space constraints.
>
> Regarding the comparison of loss functions, we compared the error in computing displacements and found that with our loss function, test error was 12% less. We also observed lower training and validation error. This has been added to the paper in Section 4.1.

---

### Meta-Review · Area_Chair_LfXU · 2021-08-16

**Recommendation:** Accept (Poster)
**Confidence:** 5

**Metareview:**

The authors propose a learning-based proprioceptive state estimator for legged robots using IMU data.  Comparison of the proposed approach to EKF and a factor-graph based estimators is presented.  As pointed out by the reviewers, the authors should discuss why they choose to ignore using the kinematic information (joint encoders) as input to the neural network.  Comparison of the method with other learning-based estimators should be presented. Failures of the proposed method should be discussed, particularly inference on out-of-distribution terrains (relearning on each terrain may not be feasible).

Having gone through the author response to reviewer comments and discussion, I am recommending an accept with poster.  I commend the authors on carefully addressing all the reviewer comments.

---

> ### Author Response · Authors · 2021-08-30
> **Response to Area Chair LfXU**
>
> We thank the Area Chair and reviewers for their careful consideration of our manuscript. We have directly addressed each of the reviewers’ comments and made several changes to the manuscript based on the reviews.
>
> ### A summary of changes to the paper is provided here:
>
> - Added learning-based baseline TLIO for all experiments.
> - Clarified a misunderstanding about training: we do not train a different model for each terrain.
> - Added discussion of limitations to learning based method.
> - Provided explanation of why we do not use kinematic information as input to the network.
>
> ### More detailed responses are provided below:
>
> **1. As pointed out by the reviewers, the authors should discuss why they choose to ignore using the kinematic information (joint encoders) as input to the neural network.**
>
> We agree with reviewer VSob that using joint state information as input to the neural network could improve the system, however we believe that this would require a significantly larger network which would simply learn the inverse kinematics. Instead, we think there could be an improvement from adding processed kinematic information such as the ground reactions forces which can be analytically computed from the joint torques. We have added discussion of this idea to the conclusion section.
>
> **2. Comparison of the method with other learning-based estimators should be presented.**
>
> We agree with several reviewers that a learning-based estimator should also be provided as a baseline. To this end, we have taken reviewer twnh’s suggestion and re-processed all lab experiments using TLIO [1]. Our method, which is based in part on TLIO, demonstrates a significant improvement because it is able to fuse information from additional sensors, such as kinematics and vision.
>
> [1] Liu, Wenxin, et al. "TLIO: Tight learned inertial odometry."
>
> **3. Failures of the proposed method should be discussed, particularly inference on out-of-distribution terrains (relearning on each terrain may not be feasible).**
>
> We appreciate the Area Chair highlighting this concern. We believe that we were unclear in the original text which led some reviewers to believe that re-training was necessary for each terrain. We have updated the text to clarify that a single model was trained from all data and used in all lab experiments. There is no re-learning for each terrain as the network can learn across the range of different terrains provided.
>
> Of course, there is still the possibility of encountering new terrain not present in the training set. While not radically different, we believe the mine experiment demonstrates  generalization from the artificial lab terrain to real world terrains with varying degrees of slipperiness and slopes. Of course the performance would likely degrade in radically different terrain such as snow and we have added a discussion of this to Section 7.1.

---

### Decision · Program_Chairs · 2021-09-13

**Decision:**

Accept (Poster)

**Comment:**

The authors propose a learning-based proprioceptive state estimator for legged robots using IMU data.  Comparison of the proposed approach to EKF and a factor-graph based estimators is presented.  As pointed out by the reviewers, the authors should discuss why they choose to ignore using the kinematic information (joint encoders) as input to the neural network.  Comparison of the method with other learning-based estimators should be presented. Failures of the proposed method should be discussed, particularly inference on out-of-distribution terrains (relearning on each terrain may not be feasible).

Having gone through the author response to reviewer comments and discussion, I am recommending an accept with poster.  I commend the authors on carefully addressing all the reviewer comments.